

# Measurements from the University of Colorado RAAVEN Uncrewed Aircraft System during ATOMIC

Gijs de Boer[1,2,3], Steven Borenstein[3], Radiance Calmer[1], Christopher Cox[2], Michael Rhodes[3], Christopher Choate[3],

Jonathan Hamilton[1,2], Jackson Osborn[2], Dale Lawrence[3], Brian Argrow[3], Janet Intrieri[2]

[1] Cooperative Institute for Research in Environmental Sciences, University of Colorado Boulder, Boulder, Colorado, USA

[2] NOAA Physical Sciences Laboratory, Boulder Colorado, USA

[3] Integrated Remote and In Situ Sensing, University of Colorado Boulder, Boulder, Colorado, USA

*Correspondence to*: Gijs de Boer (gijs.deboer@colorado.edu)

**Abstract.**  Between 24 January and 15 February 2020, small uncrewed aircraft systems (sUAS) were deployed to Morgan Lewis (Barbados) as part of the Atlantic Tradewind Ocean-Atmosphere Mesoscale Interaction Campaign

(ATOMIC), a sister project to the ElUcidating the RolE of Cloud-Circulation Coupling in ClimAte (EUREC[4]A) project.  The observations from ATOMIC and EUREC[4]A were aimed at improving our understanding of trade-wind cumulus clouds and the environmental regimes supporting them, and involved the deployment of a wide variety of observational assets, including aircraft, ships, surface-based systems and profilers.  The current manuscript describes ATOMIC observations obtained using the University of Colorado Boulder RAAVEN sUAS.  This platform collected

nearly 80 hours of data throughout the lowest kilometer of the atmosphere, sampling the near-shore environment upwind from Barbados.  Data from these platforms are publicly available through the National Oceanic and Atmospheric Administration's National Center for Environmental Intelligence (NCEI) archive.  The primary DOI for the quality-controlled dataset described in this manuscript is 10.25921/jhnd-8e58 (de Boer et al., 2021).



# 1 Introduction

The trade winds are a fundamental feature of Earth's atmospheric circulation arising from the balance of equatorward motions in large-scale overturning balanced by the Coriolis force. In the northern hemisphere, the trade winds present as northereasterlies over the sub-tropical oceans and interact with the cloud-topped marine boundary layer, which modulates the vertical transfer of energy between the sea surface and the atmosphere. For example, trade-wind clouds situated over ocean surfaces have a strong influence on how energy from the sun is transmitted, scattered and reflected

before reaching the surface in both observations (e.g., Hartmann and Short 1980) and numerical simulations (e.g., Bony and Dufresne, 2006). Additionally, these clouds help to drive larger-scale circulations that have both local and distant effects. Included is the ability of trade-wind cumulus to support mass transport out of the sub-cloud mixed layer, which in turn impacts energy exchange between the ocean surface and underlying atmosphere (e.g., Tiedtke et al., 1988; Jakob and Siebesma, 2003) Their efficiency in this regard makes them a critical component of feedbacks

sitting at the interface between the causes and responses of changes to tropical and global climate, and modeling studies have demonstrated a sensitivity of cloud structure and albedo to climate-induced changes to liquid water lapse rate, and an even greater (but opposite) sensitivity to surface-flux-induced changes to the depth and moisture content of the planetary boundary layer (Rieck et al., 2012).

In support of advancing our understanding of trade-wind cumulus clouds and the environmental regimes supporting them,

two large field campaigns were developed to simultaneously sample the tropical Atlantic Ocean east of Barbados and overlying atmosphere. Together, the EUREC[4]A (ElUcidating the RolE of Cloud-Circulation Coupling in ClimAte, Stevens et al., 2020) field study, and the Atlantic Tradewind Ocean-Atmosphere Mesoscale Interaction Campaign (ATOMIC, Zuidema et al., 2020), represent the largest-ever effort to document and understand the intraseasonal drivers of cloud properties in this region. Taking place during January and February 2020, these

campaigns build on knowledge gained through previous and ongoing observing efforts, including data from the long-term Barbados Cloud Observatory (BCO, Stevens et al, 2016) and from previous field campaigns like the Atlantic Tradewind Experiment (ATEX, Augstein et al., 1974), the Barbados Oceanographic and Meteorological Experiment (BOMEX, Holland, 1970) and the Rain In shallow Cumulus over the Ocean (RICO, Rauber et al., 2007) project. EUREC[4]A and ATOMIC saw the combined deployment of three research vessels, four manned research aircraft, and

a fleet of robotic airborne and surface vehicles which combined to provide extended regional sampling.

Through the analysis and evaluation of data collected in the field campaigns listed in the previous paragraph, there has been substantial advancement of our understanding of the complex interplay between the atmospheric state and resulting cloud cover. Specifically, a significant amount of work has gone into understanding how clouds and precipitation interact with large-scale divergence (e.g., Schubert et al., 1979), turbulent transfer of heat and moisture

between the surface and overlying atmosphere, and the thermodynamic structure of the lower atmosphere (e.g., Slingo, 1987; Albrecht, 1993). Additionally, there have been studies that aim to explain connections between cloud cover and lower atmospheric winds (e.g., Nuijens and Stevens, 2012). Interestingly, the response of clouds to environmental conditions appears to be more significant in global climate models (e.g., Nuijens et al., 2015; Bretherton 2015) than





observed during field campaigns or in high resolution simulations (e.g., Vial et al., 2017). The concept of EUREC[4]A was developed to support the unraveling of remaining mysteries associated with the interplay between clouds in the tropical trade regime, and the environment supporting their existence (Bony et al., 2017).

This publication describes the dataset collected by one of the robotic airborne vehicles deployed during ATOMIC and EUREC[4]A, the University of Colorado Integrated Remote and In Situ Sensing (IRISS) RAAVEN (Robust Autonomous Aerial Vehicle – Endurant Nimble) Uncrewed Aircraft System (UAS)[1]. The RAAVEN was deployed to document in detail the structure of the lower atmosphere, with a focus on the sub-cloud mixed-layer, throughout the length of the campaign. Specific objectives included measurement of thermodynamic and kinematic structure of the lower atmosphere, turbulent surface fluxes of heat, moisture and momentum, cloud-base mass flux, and updraft distribution in relation to cloud cover. Flights covered the time period between 24 January and 16 February 2020, and took place over the near-shore marine environment off of the northeastern coast of Barbados. In total, 39 flights were completed, resulting in a more than 77 flight-hour dataset that is documented here. After providing an overview of the RAAVEN platform and instruments it carried for ATOMIC in section 2, we provide insight into the measurement location and sampling patterns (section 3), information on data processing and quality control (section 4), information on file structure and data availability (section 5), and a summary of this observational data collection effort.

## 2 Instrument and Vehicle Descriptions

The RAAVEN UAS (Figure 1) has been operated for atmospheric science missions by the University of Colorado since early 2019. With a wingspan of 2.3 m, the airframe is a custom-manufactured model from RiteWing RC, and has been modified to meet the needs of intense atmospheric sampling measurements in a variety of environments. Specific modifications include the integration of a tail boom to enhance longitudinal stability and improve the platform's performance. The platform is equipped with a PixHawk2 autopilot system and leverages an 8S 21000 mAh Lithium Ion (Li-Ion) battery pack to offer flight times around 2.5 hours, depending on conditions and executed flight patterns. The aircraft has a top airspeed of approximately 130 km hr$^{-1}$, though operations during ATOMIC were almost exclusively conducted in the 60-70 km hr$^{-1}$ range.

For ATOMIC, the RAAVEN was equipped with an instrument suite modeled after the *miniFlux* payload co-developed by the National Oceanic and Atmospheric Administration (NOAA) and the Cooperative Institute for Research in Environmental Sciences (CIRES) and IRISS at the University of Colorado (de Boer et al., in prep.). In this configuration, the aircraft is equipped to measure a variety of atmospheric and surface properties to support evaluation of thermodynamic state, kinematic state, and turbulent fluxes of heat and momentum. This involves a collection of core instruments (see Figure 1), which include a multihole pressure probe (MHP) from Black Swift Technologies, LLC (BST), a pair of RSS421 PTH (pressure, temperature, humidity) sensors from Vaisala, Inc., a custom finewire array, developed and manufactured at the University of Colorado Boulder, a pair of Melexis MLX90614 IR

---

[1] Also known as drones, unmanned aircraft systems and remotely piloted aircraft systems



thermometers, and a VectorNav VN-300 inertial navigation system (INS). This sensor suite is logged using a custom-designed FlexLogger data logging system.

The Vaisala RSS421 sensors are identical to those used in the RD41 dropsondes deployed by the NOAA WP-3D Orion (Pincus et al., 2020) and DLR HALO aircraft during ATOMIC/ EUREC[4]A, and very similar to the RS41 radiosondes launched from a variety of platforms (Stephan et al., 2021), including the NOAA research vessel *Ronald H. Brown* (Quinn et al., 2021). This unit includes a linear resistive platinum temperature sensor that features a measurement resolution of 0.01 °C, a repeatability of 0.1 °C and a response time (as measured within the RS41

radiosonde of 0.5 s at 1000 hPa when moving at 6 m s$^{-1}$. To measure relative humidity (RH), the RSS421 leverages a thin-film capacitor with a resolution of 0.1% RH and a repeatability of 2% RH, with a temperature-dependent response time of better than 0.3 s at 20 °C (again, as measured within the RS41, with 6 m s$^{-1}$ airflow at 1000 hPa). Finally, the pressure sensor is capacitive with a silicon diaphragm, having a resolution of 0.01 hPa and a repeatability of 0.4 hPa. For ATOMIC, a pair of these sensor modules was integrated into the fuselage, about halfway between the

nose and the tail of the aircraft on the port side. The mounting angles of these two sensors were offset in order to ensure that the two sensors would have different amounts of solar exposure as the aircraft maneuvers through the atmosphere and to allow for the detection of solar heating effects since no shading is used. Additional information on atmospheric thermodynamic state is available from an E+E EE03 sensor that is integrated into the BST MHP and from a Sensiron SHT-85 sensor that is integrated in the custom finewire array. The EE-03 has a stated temperature accuracy

(at 20 °C) of 0.3 °C, while the humidity accuracy is stated to be 3% RH at 21 °C. The SHT-85 has a stated temperature accuracy of 0.1 °C (from 20-50 °C) and a repeatability of 0.08 C, while the humidity sensor has a stated accuracy of 1.5% RH and a repeatability of 0.15 % RH. Both the EE03 and the SHT-85 sensors have slower response times than the RSS421 sensor described above and are typically not used for scientific purposes unless there is a complete failure of the RSS421.


In addition to the SHT-85 sensor, the finewire array developed by the University of Colorado consists of two 5 μm diameter platinum wires extending over a 2 mm length, suspended in the free stream by supporting prongs. One wire is operated as a hotwire anemometer, with approximately 100 °C overheating compared to the ambient environmental temperature. The other wire is operated as a coldwire thermometer, with approximately 1 °C overheating relative to

the surrounding environment. These wires have thermal time constants of 0.5 ms in a 15 m s$^{-1}$ airflow regime, and support a sampling frequency of up to 800Hz. This high frequency output enables measurement of turbulent fluctuations in velocity and temperature at sampling rates up to 400Hz. A custom electronics module converts resistance change in the wires due to velocity or temperature variability in the airflow to amplified voltages that output at 800Hz. For ATOMIC, these outputs were logged at 250 Hz by the FlexLogger, which is equivalent to a 7.2 cm

minimum length scale at the RAAVEN cruise airspeed of 18 m s$^{-1}$. Time series of these recorded data are processed during post-flight analysis to calibrate the voltages recorded by the fine wire module to velocity and temperature. Additionally, these measured quantities can be fit to inertial sub-range turbulence models to wavenumber spectra over suitable time intervals, producing turbulence intensity parameters epsilon (kinetic energy dissipation rate) and $C_T^2$





(temperature structure constant). The resolution (noise floor) of these parameterizations is $2.0 \times 10^{-7}$ W kg$^{-1}$ for epsilon
and $4.5 \times 10^{-6}$ K$^2$ m$^{-2/3}$ for $C_T^2$. Resolution of the raw time series are $8.3 \times 10^{-5}$ m s$^{-1}$ for the hotwire and $1.3 \times 10^{-4}$ K for
the coldwire.

In addition to the EE-03 PTH measurements, the BST 5-hole probe provides measurements of airspeed, angle of attack
($\alpha$) and sideslip angle ($\beta$). These measurements are used together with the inertial velocities and aircraft attitude from
the VectorNav VN-300 to derive the three-components of the inertial wind (u, v, w), as discussed in section 4. The
VN-300 can be configured in a dual-Global Navigation Satellite System (GNSS) mode, under which the relative
positions of two GNSS antennae are used to calculate the platform yaw. However, this setting was not used during
the ATOMIC deployment. Under dynamic conditions, the system has a stated accuracy of 0.3 degrees in GPS-
Compass heading, 0.1 degrees in pitch and roll, 2.5 m horizontal position accuracy, 2.5 m vertical position accuracy
when integrating information from the barometric pressure sensor, and 0.05 m s$^{-1}$ accuracy in inertial velocity. Input
from the system's gyroscope, accelerometer, GNSS receiver, magnetometer and pressure sensor are filtered through
an extended Kalman Filter (EKF) to produce a navigation solution. Data from the VN-300 are logged at 50 Hz
resolution.

Finally, the RAAVEN carries a pair of Melexis MLX90614 IR thermometers, with one looking up from the top of the
aircraft, and one looking down towards the surface in level flight. These sensors are factory calibrated to work in
operational temperatures between -40 and 125 C, and to measure target brightness temperatures between -70 and 380
°C. They have a high accuracy (0.5 °C) and a measurement resolution of 0.02 °C. Unlike in the *miniFlux* sensor
system, the RAAVEN version are not stabilized to maintain a vertical orientation, meaning that the observed target is
perpendicular to the reference frame of the aircraft. This requires some care when interpreting measurement from
time periods when the aircraft is conducting pitch or rolling maneuvers. To limit the impact of surface or sky
heterogeneity, we leverage the "I" version of this sensor which has a 5-degree field of view. These sensors have a
broad passband range of 5-14 µm, meaning that while it covers the infrared atmospheric window, it is also subject to
radiation emitted by water vapor and other radiatively active gases. This means that at significant distances from a
given target (e.g., cloud; surface), atmospheric gases will influence the temperature reading. Therefore, if absolute
accuracy of brightness temperature is important, the sensor should be operated in close proximity (10s of meters or
less) to the target. However, relative contributions from different surface types or atmospheric conditions can still be
easily distinguished despite a lack of absolute calibration for extended distance sensing. Such gradient detection can
be useful for detecting surface inhomogeneities, or for understanding whether the aircraft is operating under cloud or
clear sky.

## 3 Description of measurement locations, deployment strategies and sampling

RAAVEN flights for ATOMIC were completed at Morgan Lewis, Barbados (Figure 2). Morgan Lewis is located
approximately 20 km to the NNE of Bridgetown, on the island's northeast coastline (bottom left panel of Figure 2),
and 19 km NNW of Ragged Point, where the BCO is situated. The primary operations location was Morgan Lewis



Beach (see top panel of Figure 2), with the aircraft launching from and landing on the beach surface. This location was selected to offer direct flight access to the coastline and the near-shore marine environment, and the best possible visibility of the aircraft while it conducted sampling missions over the Atlantic Ocean. The Morgan Lewis Beach is generally oriented from NNW to SSE, meaning that the prevailing wind direction was onshore, at a slight angle to the beach, offering a relatively undisturbed marine atmosphere offshore. The ocean at Morgan Lewis featured significant

surf, with swells generally around 1-3 m in height, depending on the day. On windier days, there was significant sea spray, which impacted measurements from the RAAVEN multihole pressure probe on occasion, as described in section 4. Additionally, the coast is relatively undeveloped, with limited farmland and houses. The terrain quickly rises from the ocean, with hills topping 200 m MSL approximately 1 mile inland. Vegetation along the coastline includes low-growing trees and shrubs, along with palm trees and some grassy fields.


In general, two flights were executed per day, with one flight occurring in the morning (typically around 1300 UTC, 0900 AST), and one in the afternoon (typically around 1700 UTC, 1300 AST). Some days saw an extra third flight, while others saw only one flight or none at all in order to accommodate crew rest requirements and help coordinate airspace and traffic in connection with the operation of two additional UAS in the Morgan Lewis area. Figure 3A

provides an overview of flight times throughout the ATOMIC deployment. Flights were typically a little over two hours in duration and were primarily executed in the near-shore marine environment over the Atlantic Ocean. Most of the flight time was executed in a racetrack pattern that was oriented parallel to shore, though profiling was completed using a loiter circle, as was the lowest flight level at approximately 20 m above the ocean surface. Additionally, most flights saw at least one leg oriented at 90 degrees relative to shore.


The primary flight pattern executed by the RAAVEN is shown in Figure 3B. This pattern included a launch from the beach, followed by a couple of nearby, low altitude orbits to verify system functionality. Once the system was ready to continue its mission, the aircraft was first sent to conduct an orbital profile (225 m diameter) between 20 and 1000 m altitude, assuming that cloud cover allowed for a profile to this altitude. If cloud cover was extensive, then the

profile was capped at cloud base. This profile was included to obtain information on lower atmospheric thermodynamic and dynamic structure, extending from the surface, through the boundary layer, and into the cloud layer capping the boundary layer. After reaching the top of the profile, the aircraft would descend to the boundary layer top ("cloud base"), as detected using data from a ceilometer at the BCO, situated approximately 19 km to the SE of Morgan Lewis, and by visual confirmation of the aircraft being below the clouds. Once at cloud base, the aircraft

would execute a horizontal "racetrack" pattern spanning approximately 2.5-3 km and oriented approximately parallel the shoreline for an extended flight period (typically 20-40 minutes) to collect statistics on the spatial variability of thermodynamic and kinematic properties of the cloud base environment. Specific measurement targets for this flight leg included the cloud-base mass flux, vertical velocity distribution and eddy length scale at cloud base, turbulent fluxes of heat and momentum through the top of the atmospheric boundary layer, and cloud horizontal extent.

Following this, the aircraft would execute additional extended racetrack sampling legs at 400 m and 200 m MSL to collect statistics to evaluate atmospheric properties, including turbulent fluxes of heat and momentum, and vertical





velocity distributions and eddy length scale in the center of the boundary layer, at these two levels. During these racetrack flight patterns, the aircraft was typically additionally operated to conduct one racetrack that was oriented perpendicular to the others to help with calibration of wind measurements. Finally, the aircraft would descend to

approximately 20 m MSL and conduct 20 minutes of circular orbiting at this low altitude to measure quantities related to the surface layer, the underlying ocean, surface-atmosphere exchange and the turbulent structure of the lower atmosphere. This orbit was positioned to be outside of the near-shore surf zone to avoid the influence of breaking waves on these measurements. Following this low-altitude leg, the aircraft would return to the beach for landing.

In addition to this primary flight pattern, the RAAVEN was also operated to conduct a more complex pattern designed to help with the calibration of winds. This pattern was typically carried out at 100 m MSL, and included a series of three racetrack patterns (~2 km) oriented in directions approximately offset by 120 degrees to create a star pattern. This offered a variety of flight directions relative to the very steady (both in speed and direction) trade-winds that the aircraft was operating in. This pattern was specifically designed to assist with evaluating any biases in the wind

measurement system, and is fashioned after flight patterns recommended in section 6 of van den Kroonenberg et al. (2008).

Over the course of the campaign, the completed flights sampled a variety of altitudes, with enhanced sampling at 20, 200 and 400 m, and near cloud base (variable altitude). This resulted in the distribution of sampled altitudes shown

in Figure 3C. These levels of enhanced sampling are meant to support the analysis of statistics at a given level, as may be required for evaluating turbulent transport of heat and momentum, and as needed for understanding wind and thermodynamic variance. Additionally, the cloud-base legs are meant to support the evaluation of the vertical velocity structure associated with the cumulus clouds present in the tropical trades.

### 4 Data processing and quality control

Data collected by the RAAVEN during ATOMIC were logged at a variety of different rates, depending on the sensor. The custom finewire sensor was logged at 250 Hz, the fastest rate of all of the sensors. The BST MHP was logged at 100 Hz, the VectorNav VN-300 at 50 Hz, the Melexis IR sensors and variables related to finewire status were logged at 20 Hz, while data collected from the PixHawk autopilot and RSS421 sensors were logged at 5 Hz. For each logging event, a sample time from the clock on the Flexlogger system is recorded, allowing for post-collection time alignment.

These sample times, along with artificial 5, 20, 50, 100, and 250 Hz clocks spanning the sample time that the VN-300 first acquired GPS lock to the last recorded sample time for the VN-300, are used to align the times of the different variables to a set of common clocks, primarily through one-dimensional linear interpolation. One exception to the linear interpolation is the yaw estimate, which is circular in nature (ranging between -180 and 180 degrees), which uses a "nearest" interpolation where the value at the nearest time stamp is used. During this interpolation process, a

limited number of points sharing a common sample time with another point are removed from the record. Once these time variables are established, a *base_time* variable is established using the 250 Hz time stamp, and offsets from *base_time* are then calculated for all different logging resolutions.



Once the initial time stamps are established, the resampled dataset includes a variety of derived and measured
quantities. Aircraft position, including latitude and longitude, are measured by the VN-300. The aircraft altitude is
derived using a combination of various inputs. While the aircraft carries sensors to measure both GPS and pressure
altitude, neither of these outputs can be used reliably as the flight altitude. The pressure altitude is subject to drift over
the duration of a single flight, potentially resulting in values at landing that are higher or lower than those at take-off.
Similarly, the accuracy of the GPS altitude is insufficient to accurately capture the vertical position of the aircraft.
Additionally, the VN-300 was not calibrated specifically for this location, resulting in a significant negative bias (~50
m) for its reported altitude. To calculate a true altitude, a combination of the autopilot altitude, VN-300 altitude, and
VN-300 pressure are used. First, a *flight_flag* variable is computed using airspeed and altitude information from the
autopilot. Any data points with airspeed exceeding 10 m/s and an altitude exceeding 5 m is flagged as a time when
the aircraft is flying (*flight_flag*=1). The point at 200 samples (4 seconds) prior to the first point in the record where
the aircraft is deemed to be flying is recorded as the initial take-off index, while the data point at 200 samples (4
seconds) after the last point in the record where the aircraft is deemed to be flying is recorded as the landing index.
The difference between the autopilot altitude at launch and at landing is added into the flight record on a timestep-by-
timestep basis, to correct for temporal drift in pressure. A linear fit is then calculated to relate the VN-300 pressure
and the difference between the VN-300 reported altitude and the autopilot reported altitude. This pressure-dependent
altitude correction is then applied to the VN-300-reported altitude to derive a final altitude.

Estimation of winds from fixed-wing aircraft requires a combination of a variety of measurements related to airspeed,
aircraft motion, and airflow over the aircraft (see van den Kroonenberg et al., 2008). These measurements need to be
of sufficient quality, and any angular offsets and logging delays need to be considered and removed. This includes
the removal of effects related to biases in the reported aircraft true airspeed (TAS), application of a time-lag correction
to account for potential delays separating the VN-300 reported GPS velocities and the airspeed, and correcting for
possible angular offsets between the VN-300 and multihole pressure probe. For the RAAVEN system, biases in TAS
are found to have the largest impact on wind speed derivation, while the time-lag correction was found to have the
smallest impact. All of these potential sources of error are corrected for using an optimization technique, where small
adjustments are made to the individual parameters and the combination that results in the wind solution with the
smallest overall variance is selected as the correct winds.

For the RAAVEN ATOMIC dataset, TAS is calculated using measurements from the MHP and RSS421 probe. Using
equation 1 from Brown et al. (1983), TAS can be calculated as:

$$TAS_i = \sqrt{\frac{2\bar{q}}{\rho}} \tag{1}$$

where $\rho$ is the density of air, calculated as:

$$\rho = \frac{p_s}{R_d T} \tag{2}$$

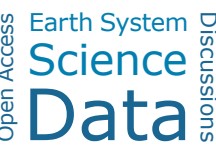

where $p_s$ is the static pressure (Pa) reported from the MHP, T is the air temperature (K) reported by the RSS421, and $R_d$ is the specific gas constant for dry air, equal to 287.058 J kg⁻¹ K⁻¹. $\bar{q}$ is defined as:

$$\bar{q} = \frac{p_0}{1 - \frac{9}{4}\sin^2 \theta_a} \tag{3}$$

where $\sin^2 \theta_a$ is the total aerodynamic angle of the MHP, calculated using the angle of attack (α) and sideslip angle (β) reported by the MHP.

From testing in a temperature chamber, it was found that the pressure sensors used in this version of the MHP had non-linear temperature dependencies. These dependencies were, unfortunately, not found to be consistent between 285 the five sensors in the MHP. Therefore, after an initial calculation of the TAS using the technique above, an additional temperature-dependent correction was applied to ensure that an artificial alteration of TAS with altitude was not present. Using a linear regression, for this particular probe it was found that TAS needed to be corrected as follows:

$$TAS = TAS_i - \big(.1193 \text{ x } (27.25 - T_C)\big) \tag{4}$$


where $T_C$ is the temperature from the RSS421 sensor in °C. Efforts are currently being undertaken to improve the MHP measurements by including a temperature-dependent correction of the pressure data, as well as a full wind-tunnel-based calibration of the MHP itself to evaluate the factory-supplied calibration coefficients.

Development of thermodynamic measurements from the RAAVEN included multiple processing steps. First, the data from the two RSS421 sensors are averaged to attempt to reduce the influence of any solar exposure of the sensors. After review of the temperatures from these sensors as a function of yaw angle, we did not see evidence of solar influence on the measurement. Nevertheless, because the sensors were oriented at different angles relative to the airframe it is unlikely that they were both impacted by solar exposure at the same time, and there is further opportunity 300 to reduce any solar influence. We do this by taking the average value of the two sensors as the ambient temperature. Typically, the two sensors vary by less than 0.2 °C (see Figure 4). After this temperature time series has been developed, a temperature calibration is derived for the coldwire data. This is done by developing a linear fit to the relationship between the coldwire voltage and the temperature measured by the RSS421 sensor. As is the case with the RSS421 temperature measurements, the RSS421 relative humidity values are also averaged. Typically, these 305 measurements agree to within 2%. The relative humidity measurements did undergo one additional correction for the first two flight days due to the fact that the sensors were not conditioned prior to flight as required, resulting in measurements that have a dry bias. Fortunately, we were conducting flights around the same time that a radiosonde was launched from the BCO, and we were able to apply a linear correction to the data for these first two flights, taking the form of:





$$RH_{adj} = RH_{orig} + 5.5 + \frac{Z}{140} \tag{5}$$


Where $RH_{adj}$ is the corrected relative humidity, $RH_{orig}$ is the measured relative humidity, and $Z$ is the altitude of the aircraft above the surface in meters. This correction assumes that over the two-day period prior to the sensors being conditioned, the bias is constant. Additionally, it assumes that there is a temperature dependence associated with the correction which is accounted for with the height-dependent scaling factor since the flights in question took place within the mixed layer.


All of the measured quantities have data quality flags associated with them. For the RSS421-derived temperature, the flag is set to zero for good data, and set to one for times when any of the following occur: The absolute value of the difference between the temperature from either individual sensor and the output temperature is greater than 0.5 °C, the absolute value of the difference between the output temperature and the temperature from the EE-03 sensor on the MHP exceeds 5 °C, the recorded error flag of either RSS421 sensor is active, or the aircraft is not flying. For the RH measurement from the RSS421, a similar set of criteria are used to activate the data quality flag, except the limits are set to be 5% between RSS421 sensors, and 15% between the output RH value and the MHP-provided RH value. The coldwire temperature data quality flag is activated when the difference between the coldwire temperature and either of the RSS421 temperatures exceeds 0.6 °C, when the absolute value of the difference between the coldwire temperature and the MHP temperature exceeds 2 °C, when the coldwire voltage is observed to fall outside of the 0-4 V analog range, or when the aircraft is not flying. Finally, the pressure quality control flag for the pressure measurement from the VN-300 is activated if the absolute value of the difference between the reported VN-300 static pressure and that measured by either of the RSS421 sensors exceeds 2 hPa. The RSS421 pressure measurements are not used because they are believed to be biased low due to the airflow passing over their location on the aircraft.




Finally, we include a 3-stage flag for the wind measurements, which is set to 0 (good data), 1 (suspect data) or 2 (bad data). Data are determined to be bad if any of the following conditions were met:

- The measured angle of attack or sideslip exceeds 20 degrees, with values between 10-20 degrees are flagged as "suspect"


- The true airspeed (TAS) is below 10 m s[-1]
- Any of the MHP ports are deemed to be blocked, as determined by the differential pressure value for any of the sensors falling below -100 Pa
- The moving window variance of the MHP-derived TAS over 40 seconds is less than 5


- The aircraft is not flying
- The difference between the MHP TAS and that from the Pitot probe is greater than 5 m s[-1]

To evaluate the accuracy of the RAAVEN observations, we completed a comparison with measurements collected by radiosondes launched from the BCO (Stephan et al., 2020). Results from this comparison are shown in Figure 5. This comparison includes any periods where the RAAVEN sampling time and that of the radiosonde are less than one hour apart. While the radiosondes were launched approximately 20 km to the southeast, we believe that the air sampled by




both systems is largely representative of the marine boundary layer, given that wind directions were always between ESE-ENE. Both datasets are averaged onto a common altitude grid which includes points between the surface and 1060 m MSL, in 20 m altitude steps. Figure 5 shows how well the RAAVEN observations match those from the radiosondes, and demonstrate the amount of variability in individual variables, with air pressure and wind direction typically varying very little, temperature a bit more, and wind speed and specific humidity having the largest amount of small-scale variability. However, for all measurements the observations fall around the one-to-one line, supporting the idea that the RAAVEN measurements are providing an accurate depiction of the lower atmosphere, relative to radiosondes launched from the BCO.

Finally, to allow the data user to better understand the aircraft's flight state and to allow for improved selection of specific phases of flight, we included two additional flags in the datastream. These include the "Flight_Flag" introduced previously, as well as a "Flight_State" flag. The "Flight_State" flag includes information on whether the aircraft is in flight (1) or not (0) in the hundreds place, whether the aircraft is descending (0), level (1), or ascending (2) in the tens place, and whether the aircraft is flying straight (0) or is turning (1). For example, if a data user wanted to find straight, level legs in flight, they would search for data with "Flight_State" equal to 110. These flags are derived from information from a combination of sensors, including the altitude variable described above, the aircraft yaw, and the "Flight_Flag" variable described earlier on in this paragraph.

Figure 6 illustrates distributions of several key parameters across the entire ATOMIC campaign. These distributions include "in-flight" data only and reveal the relative consistency of the structure of the lower atmosphere in the trade-wind regime. Despite covering nearly a month of time, these observations have relatively narrow distributions in most parameters, with the majority of the variability being a function of altitude, rather than a function of synoptic variability. Peaks in the air temperature and pressure data are the result of the aircraft spending significant amounts of time at a series of levels between cloud base and the surface.

## 5    Data Availability and File Structure

ATOMIC RAAVEN data are available for public download at the NOAA National Center for Environmental Intelligence (https://doi.org/10.25921/jhnd-8e58, last accessed 23 April 2021). Files are provided in NetCDF format, with the filename convention ATOMIC_CU-RAAVEN_YYYYMMDD_HHMMSS_B1.nc, where the date reflects the power-on time for the aircraft. In total, there are 39 files totaling 703 MB. Files described in this manuscript and accessible using the above DOI are quality controlled and labeled as "B1". The raw data (A1 files) are not currently archived, but are stored on local backed-up disk storage, and can be accessed if necessary.

## 6 Summary

Between 24 January and 15 February 2020, the University of Colorado and NOAA Physical Sciences Laboratory deployed the RAAVEN small uncrewed aircraft platform to Morgan Lewis, Barbados as part of the ATOMIC field





campaign. Over the course of three weeks, the team collected nearly 80 hours of flight data over 39 flights. These flights were generally conducted at a 2x daily frequency, with one morning and one afternoon flight. These flights provide extensive observations of the thermodynamic and kinematic structure of the marine boundary layer (between surface and 1000 m MSL) in the tropical trades. It is believed that these data offer high-resolution insight not readily

achieved with other observing techniques. Due to the slow flight speed of the RAAVEN (18 m s$^{-1}$) and the platform's extensive endurance (2+ hours), observations collected span length scales from approximately 5 cm to 130 km.

Data were processed and quality-controlled to provide consistent data at 10 Hz temporal resolution. This includes the correction of wind, temperature and humidity data. A comparison with radiosondes launched approximately 20 km

away was conducted, and reveals good agreement between the RAAVEN observations and those from the radiosondes. It is believed that these data offer new perspectives that can be leveraged to gain better understanding of the interactions between the surface and cloud layer, including turbulence, turbulent transport, internal variability in thermodynamic and kinematic variables, and turbulent surface fluxes. In conjunction with the other observations collected during ATOMIC and EUREC[4]A, these data should provide the basis for a variety of studies over the coming

years.

**Author Contributions.**

All co-authors contributed to the development of this dataset. GB coordinated the collection of the data and deployed to Barbados for ATOMIC, along with SB, RC, MR, C. Choate, JH and JI. C. Cox and RC helped to support the

analysis, preparation and quality control of these data. JO, DL and BA contributed to the preparation of the vehicles and other items for field deployment. GB led the development of the manuscript, with all authors contributing to editing the resulting paper.

**Competing Interests.**

GB is a co-editor for the special issue in which this manuscript appears. Additionally, GB works as a consultant for Black Swift Technologies, who manufacture the multi-hole pressure probe used in the collection of this dataset.

**Special issue statement.**

This article is part of the special issue "Elucidating the role of clouds–circulation coupling in climate: datasets from

the 2020 (EUREC[4]A) field campaign". The observations presented here were collected under the ATOMIC (Atlantic Tradewind Ocean-Atmosphere Mesoscale Interaction Campaign) project, which took place under the broader EUREC[4]A umbrella.

**Acknowledgements**.

We acknowledge the significant planning that went into the development of the ATOMIC and EUREC[4]A projects, and would like to specifically thank Drs. Christopher Fairall, Bjorn Stevens, Sandrine Bony and David Farrell for their efforts in planning and executing these major campaigns. We would specifically like to thank Dr. Farrell for his





assistance in working with local authorities to obtain the import and flight permissions required to conduct these flights. Furthermore, we would like to acknowledge Dr. Gregory Roberts and Mr. Christophe Mazel for their collaboration and for their coordination of the RPAS activities occurring from Morgan Lewis, and allowing us to store equipment in their containers throughout the field campaign. Funding for this work was provided by the NOAA UAS Program Office, the NOAA Climate Program Office and the NOAA Physical Sciences Laboratory.




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

Zuidema, P. and co-authors: The Atlantic Tradewind Ocean–Atmosphere Mesoscale Interaction Campaign (ATOMIC): Plumbing the Ocean Depths and Probing the Atmosphere, *Bull. Amer. Meteor. Soc.*, in prep.

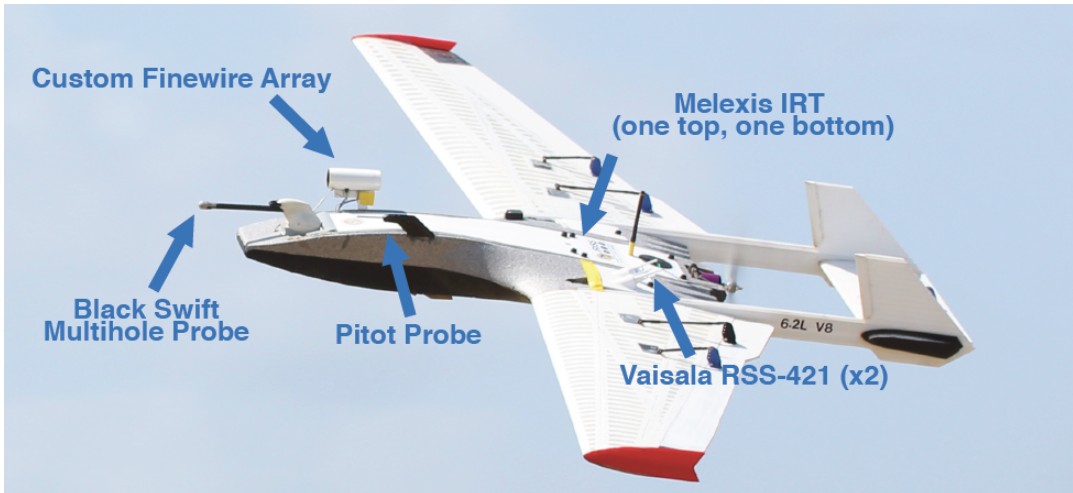

**Figure 1:** Figure showing a picture of the University of Colorado RAAVEN UAS, as operated in Barbados for ATOMIC, with labels pointing out the locations of the individual sensors.


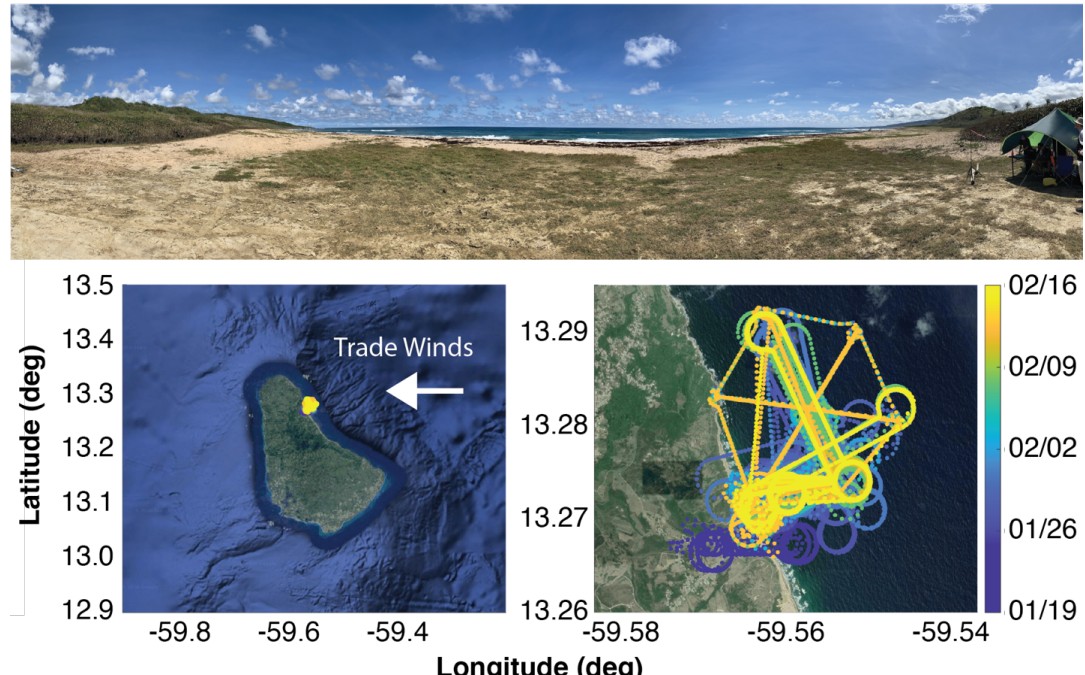

**Figure 2:** Morgan Lewis beach (top) as seen from our operations site. A map of Barbados, showing the location of
RAAVEN flights, relative to the rest of the island (bottom left). Details on the flight tracks completed, color-coded
by date (bottom right). Background maps are © GoogleMaps 2021, downloaded through their API.


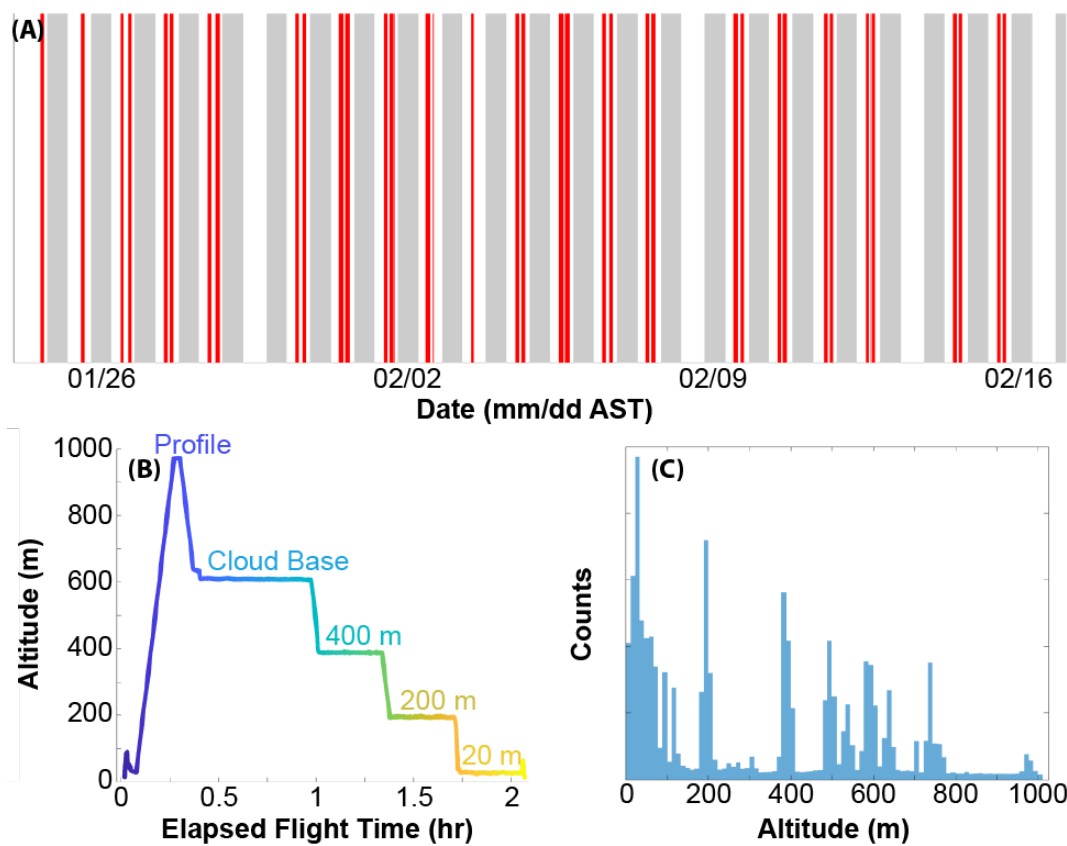

**Figure 3:** A figure illustrating information on the completed flights. The top panel (A) shows RAAVEN flight times (red bars) relative to nautical twilight (grey bars). The bottom left-hand panel (B) shows an example of the most commonly-flown pattern, including a profile to 1 km, and then extended sampling legs at cloud base, 400 m, 200 m and 20 m (colors represent time into the flight). The bottom right-hand panel (C) shows a histogram of flight altitudes covered by the RAAVEN during ATOMIC.



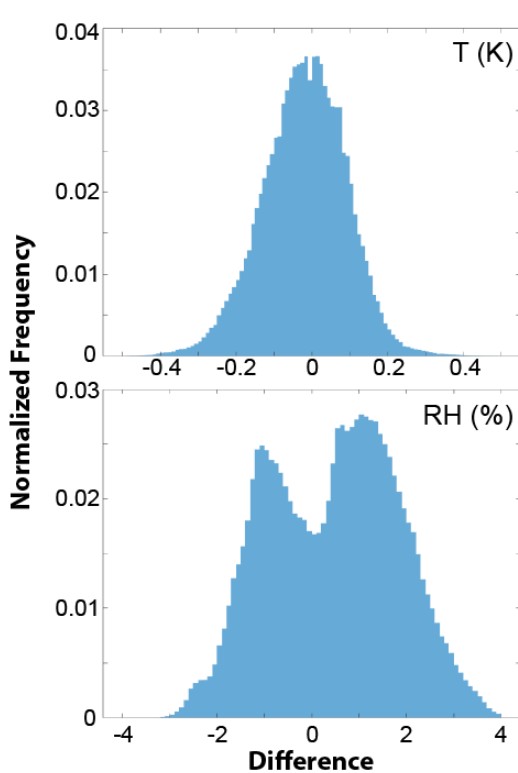

**Figure 4:** Distributions of differences in the RSS421 measured temperatures (top) and relative humidities (bottom) for all times the aircraft was in flight during the ATOMIC campaign.



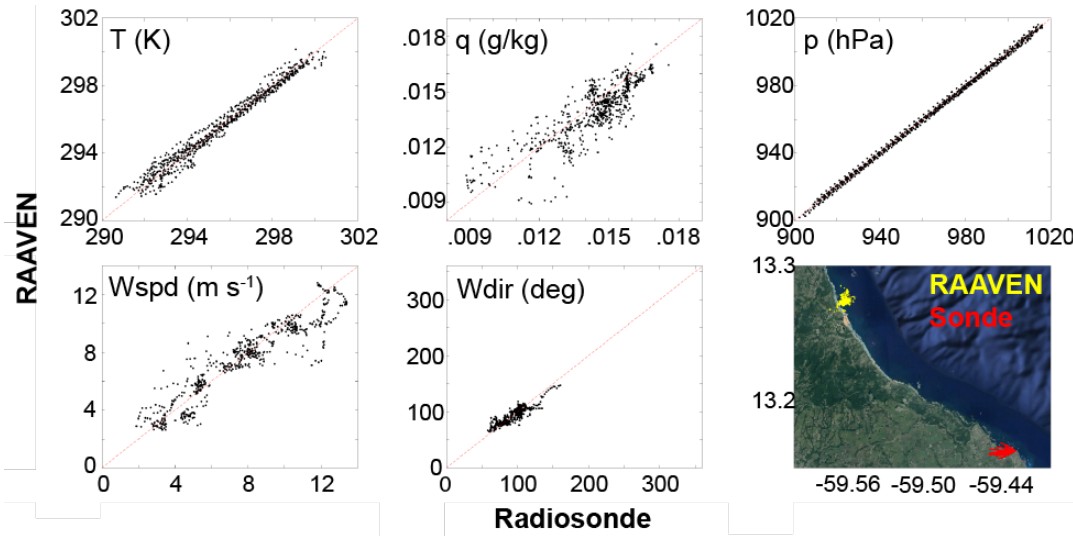

**Figure 5:** A comparison between radiosonde observations and those from the RAAVEN. The lower righthand panel shows the positions of the RAAVEN and radiosonde during times of shared altitude used for this comparison. Quantities compared include air temperature, specific humidity and air pressure (top row, left to right), and wind speed and direction (lower row, left and center). Background maps are © GoogleMaps 2021, downloaded through their API.

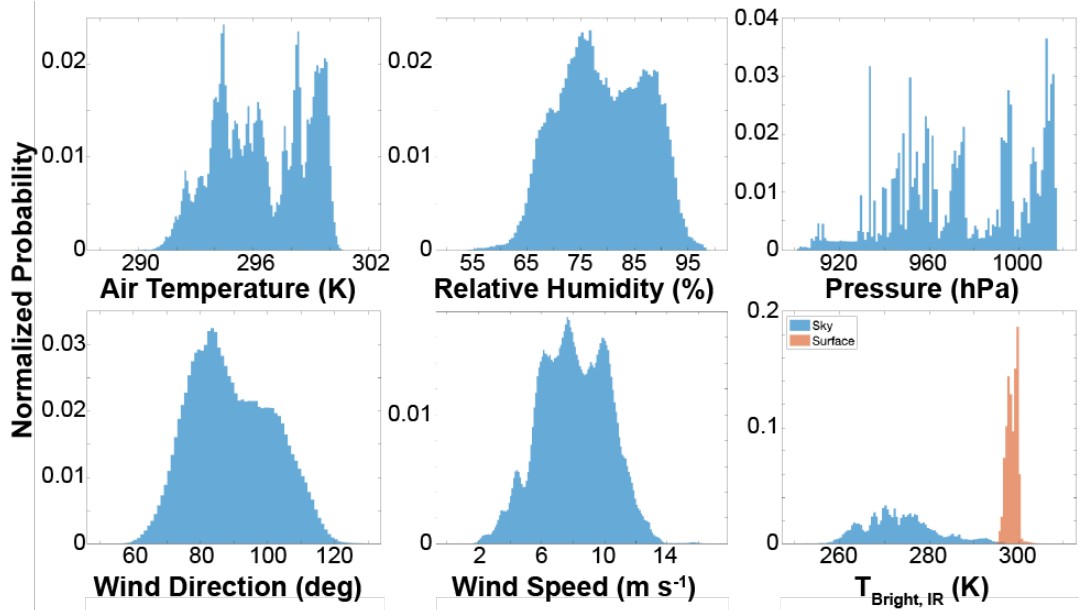

**Figure 6:** Histograms of the primary measurements from RAAVEN, collected while in flight, over all flights conducted during ATOMIC.