# Peer review of "Measurements from the University of Colorado RAAVEN Uncrewed Aircraft System during ATOMIC"

_Earth System Science Data, 2021_

## Author Response (AR1)

**Reviewer 1:**

General comments:

This paper provides promising 80 hours of data for environmental research to improve the understanding of the interaction between the surface and cloud layer. The authors presented a very useful payload for the atmospheric study and demonstrated new perspectives the data offer. In addition, the data is high quality, and the manuscript is well-written. Thus, I recommend the publication with several minor suggestions.

**We appreciate the feedback provided by the reviewer and the comments provided to improve the overall quality of the manuscript.**

Specific comments:

P4, lines 95-115, The Vaisala sensors resolution and repeatability are not under the flight condition (60-70 km/hr). What is the uncertainty for the temperature and RH data using this sensor? Will you please comment on the aerodynamic effect of the flow distortion caused by the aircraft and the sensor on the measured temperature? Or how much different will it be related to the static air temperature? Figure 6 shows the air temperature. Is it static air temperature?

**The temperature shown in Figure 6 is total air temperature. Assuming ideal gas qualities, for the low speeds at which the RAAVEN flies (17 m/s cruise speed), the difference between total and static temperature is quite small (~ 0.14 degrees for the ATOMIC temperature range). While the reviewer is correct in stating that the resolution and repeatability are not necessarily for the flight velocity, the velocities of the RAAVEN are not that different than what these numbers are cited to include, as the RD-41 spec sheet states an anticipated descent rate of 11 m/s at sea level. We do not believe that the resolution and repeatability are going to be impacted much by additional airspeed, though the unit response times will impact the spatial scales of variability that will be captured with a slightly faster moving platform. In this light, the repeatability can be interpreted as the measurement uncertainty, with the repeatability representing the standard deviation of differences between two successive repeated calibrations (k=2 confidence level). Finally, while aerodynamic effects have not been fully evaluated, we have conducted multiple trials where the RAAVEN was flown alongside instrumented towers, including at the US Department of Energy Atmospheric Radiation Measurement (ARM) program facility in Oklahoma, the National Renewable Energy Laboratory (NREL) Flatirons Campus in Colorado, and the National Center for Atmospheric Research (NCAR) Marshall site in Colorado, and compared to radiosonde data for closely-coordinated profiles. These flight tests have**

**demonstrated close agreement between the RSS-421 temperature measurements and do not show large impacts from aerodynamic effects or heating of the aircraft body (see Fig. 1 below). These comparisons have yet to be published (currently in prep) and are therefore not discussed and/or cited in the current manuscript.**

[Figure]

**Figure 1: Comparison between UAS-measured temperatures and those from RS-41 radiosondes launched at the US Department of Energy ARM facility. Included are data from the CU RAAVEN (yellow), Black Swift S0 (black), OU Coptersonde (maroon), UNL M600 (light red) and UNL Meteodrone (darker red).**

P4, line 124, the SHT-85 sensor was logged at 250 Hz. Does this sampling rate reflect the actual measured value, or is it based on the electronics response time? Do you have any spectral analysis or reference to confirm the actual response rate of the sensor?

**The response rate of the SHT-85 sensor is substantially lower than 250 Hz. We logged this sensor at this rate because it is part of the finewire array and we wanted high temporal resolution data from the 6 micron hot and coldwires in that array.**

P5, line 139-141, Are the system accuracies under the 50 Hz sampling resolution? Or 1 Hz sampling resolution? Data from the VN-300 were logged at 50 Hz resolution. How many variables recorded are real measured values? It will be great to share more experience on how to use VN-300 under higher time resolution reliably.

**The data logged from the VN-300 are from an INS filter, so technically none of the values are "real measured values". We're not sure we understand the reviewer's question about the accuracies and the difference between 50 and 1 Hz data. The Vector-Nav spec sheet does not provide information on the impact of logging rates on system accuracies. The sensors that feed the Kalman filter have a bandwidth of over 250 Hz.**

P9, section 4, what are the error propagation to the TAS estimation and the wind estimation?

**This is a complex question, given the processing that is applied to the current dataset. Answering the question directly, based on wind tunnel testing, the MHP is shown to have random statistical variability of around 0.25 m/s (roughly +/- 1.5% relative to cruise speed if it is assumed that the reported TAS is within 0.25 m/s of truth). Carrying this number forward to estimate the systematic error this results in a 6% mean error in wind speed (0.6%/9% 10th/90th percentiles– see Figure 2 below). Having said this, part of the minimization approach involves making small changes to the TAS to reduce possible biases. This was possible for the ATOMIC campaign due to the very steady nature of the tropical trade winds. Because these winds are so steady, it is assumed that aircraft flight direction should not matter, and that any steps observed in the measured winds associated with flight directional changes are the result of probe miscalibration or alignment issues. Therefore, the multivariate optimization technique makes small adjustments to TAS to minimize these steps across the flight. Figure 3 below shows a timeseries of the "optimized" wind (blue) as well as the wind when calculated with a 3% error in TAS (red) – note the steps shown in the red wind that are associated with the aircraft turning at the end of its racetrack legs. Given this approach, any biases in TAS have been removed to the level of the 3% statistical error referenced above.**

[Figure]

**Figure 2: A histogram of the percentage error in wind speed resulting from a TAS offset of 3%. The dashed black lines represent the 10th and 90th percentiles, while the solid black line represents the mean.**

[Figure]

**Figure 3: A timeseries of wind speed from the RAAVEN for a portion of flight 8 from ATOMIC. The blue line represents the reported value, while the red line represents the wind speed calculated with a TAS offset of 3%.**

**Reviewer 2:**

The report and corresponding dataset provide a unique and interesting set of observations in support of the ATOMIC campaign. The authors do a good job of describing their systems and quality control process, and provide intercomparison results which provide confidence in the quality of the data available. The report itself is well written and mostly complete. However, I do have some small comments that should be addressed before publication:

**We appreciate the time taken by the reviewer to provide feedback on this paper.**

- epsilon is usually provided as the greek symbol and not spelled out. If the authors do not wish to use greek symbols, for the limited use of 'epsilon', it would seem to me that using the phrase `turbulent kinetic energy dissipation rate' would be sufficiently descriptive

**Thank you for pointing this out. We have inserted the Greek symbol ($\varepsilon$) to replace the spelled-out version.**

- in equation (3) $p_0$ is not defined. Presumably it is the dynamic pressure measured by the central hole of the MHP?

**That is correct. This is derived from equation 1 in Brown et al. (1983), as cited in the text. We have included a definition for $p_0$ in the manuscript.**

- When referring to the optimization of the wind estimates from the MHP, the authors state the 'small adjustments are made to the individual parameters'. Which parameters are adjusted, and is there a rationale behind the adjustment? Or is this a multivariate optimization process?

**Good question, and we apologize for not being clearer in the text. This is a multi-variate optimization process that includes adjustments to TAS, pitch and yaw to account for probe miscalibration and angular offsets between the IMU and the multihole probe. The range of values for the adjustment includes 10% for TAS and 10 degrees for yaw and 3 degrees for pitch. Adjustments for the Euler angles are made with a full rotation of the coordinate system. We have updated the text to outline these details more clearly.**

- The authors provide nice detail about the sensors and which data is provided from the sensors. However, it would be convenient for those accessing the data if there were a summary table listing the variables provided in the dataset, the units provided, and possibly the sensor(s) used to produce that variable.

This information is largely found in the data files themselves. Within the NetCDF file, each variable is named, where possible using CF conventions, and information on units is included. Additionally, for most variables, information is provided on which sensor that measurement comes from. Nonetheless, we have added the requested table to the manuscript to provide readers with easy access to this information.